

**Title:** Methanol and Isoprene Emissions from the Fast Growing Tropical Pioneer Species *Vismia*
*guianensis* (Aubl.) Pers. (Hypericaceae) in the central Amazon Forest
**Authors:** Kolby J. Jardine[*1], Angela B. Jardine[2], Vinicius F. Souza[2], Vilany Carneiro[2], Joao V.
Ceron[2], Bruno O. Gimenez[2], Cilene P. Soares[2], Flavia M. Durgante[2], Niro Higuchi[2], Antonio O.
Manzi[2], José F. C. Gonçalves[2], Sabrina Garcia[2], Scot T. Martin[3], Raquel F. Zorzanelli[2], Luani R.
Piva[2], Jeff Q. Chambers[1,4]
[1]Climate and Ecosystem Sciences Division, Earth Science Division, Lawrence Berkeley National
Laboratory, One Cyclotron Rd, building 64-241, Berkeley, CA 94720, USA
[2]National Institute for Amazon Research (INPA), Ave. Andre Araujo 2936, Campus II, Building LBA,
Manaus, AM 69.080-97, Brazil
[3]School of Engineering and Applied Sciences and Department of Earth and Planetary Sciences, Harvard
University, Cambridge, Massachusetts, USA
[4]Department of Geography, University of California Berkeley, 507 McCone Hall #4740, Berkeley, CA,
94720, USA
[*]**Corresponding author:** Kolby Jardine *Climate Science Department, Earth Science Division,*
*Lawrence Berkeley National Laboratory, One Cyclotron Rd, building 64-241, Berkeley, CA 94720, USA,*
*email* (kjjardine@lbl.gov)
Observations and Modeling of the Green Ocean Amazon (GoAmazon 2014/5) inter-journal
special issue (ACP/AMT/GI/GMD)
**Key Points**
• High light-dependent isoprene emissions were observed from mature *V. guianensis*
leaves in the central Amazon
• As predicted by energetic models, isoprene emission increased non-linearly with net
photosynthesis
• High leaf temperatures resulted in the classic uncoupling of net photosynthesis from
isoprene emissions
• Leaf phenology differentially controls methanol and isoprene emissions





**Abstract**

Isoprene (Is) emissions by plants represent a loss of carbon and energy resources leading to the initial hypothesis that fast growing pioneer species in secondary tropical forests allocate carbon primarily to growth at the expense of isoprenoid defenses. In this study, we quantified leaf isoprene and methanol emissions from the abundant pantropical pioneer tree species *Vismia guianensis* and ambient isoprene concentrations above a diverse secondary forest in the central Amazon. As photosynthetically active radiation (PAR) was varied (0 to 3,000 µmol m$^{-2}$ s$^{-1}$) under standard leaf temperature (30 °C), isoprene emissions from *V. guianensis* increased without saturation up to 80 nmol m$^{-2}$ s$^{-1}$. A non-linear increase in isoprene emissions with respect to net photosynthesis (Pn) resulted with the fraction of Pn dedicated to isoprene emissions increasing with light intensity (up to 2% of Pn). Emission responses to temperature under standard light conditions (PAR of 1,000 µmol m$^{-2}$ s$^{-1}$) resulted in the classic uncoupling of isoprene emissions ($T_{opt,iso} > 40$ °C) from net photosynthesis ($T_{opt, Pn} = 30.0$-$32.5$ °C) with up to 7% of Pn emitted as isoprene at 40 °C. Under standard environmental conditions of PAR and leaf temperature, young *V. guianensis* leaves showed high methanol emissions, low Pn, and low isoprene emissions. In contrast, mature leaves showed high Pn, high isoprene emissions, and low methanol emissions, highlighting the differential control of leaf phenology over methanol and isoprene emissions. High daytime ambient isoprene concentrations (11 ppbv) were observed above a secondary Amazon rainforest suggesting that isoprene emissions are common among neotropical pioneer species. The results are not consistent with the initial hypothesis and support a functional role of methanol during leaf expansion and the establishment of photosynthetic machinery, and a protective role of isoprene for photosynthesis during high temperature extremes regularly experienced in secondary rainforest ecosystems.

**Keywords:** Pioneer species, tropical forest, growth and defense, volatile isoprenoids

## 1. Introduction

Due to its vast territorial expansion, high species diversity, and long growing season, the Amazon forest in South America is responsible for an estimated 15% of global terrestrial photosynthesis (Malhi et al., 2008). However, increased deforestation, degradation and natural disturbances have changed this scenario, exerting strong control on the evolution of atmospheric $CO_2$ (Pan et al., 2011;Malhi et al., 2008). A recent analysis of biomass dynamics revealed a long-term trend of increased mortality-driven shortening of carbon residence times in the Amazon forest (Brienen et al., 2015). This effect has been attributed to increased climate variability, as recurrent drought episodes occurred in the region (Phillips et al., 2009;Lewis et al., 2011). Amazon carbon sink suppression during the intense drought period in 2005 was associated with a decrease in biomass gain and increased vegetation mortality (Phillips et al., 2009). Changes in forest turnover rate can directly affect forest composition and structure; the creation of forest gaps leads to the release of suppressed trees and increased pioneer species recruitment rates





(Bugmann, 2001). Tropical forest regrowth has been identified as a strong terrestrial carbon sink
that can partly counterbalance carbon losses by deforestation and forest degradation (Pan et al.,
2011). If tropical forests are becoming more dynamic, gap-phase processes can therefore play a
more central role in determining carbon residence times, which have been described as the
largest uncertainty in terrestrial vegetation responses to climate and elevated $CO_2$ (Friend et al.,
73 2014).

The classic Neotropical pioneer genera *Vismia* and *Cecropia* dominate large rainforest
disturbance gaps in the Amazon Basin (Chambers et al., 2009) where they help accelerate the
regeneration of secondary forests by influencing forest successional pathways (Uhl et al.,
1988;Vieira et al., 2003;Zalamea et al., 2008). Their success in secondary forests is related to
their ability to maintain high rates of net photosynthesis (Pn) and growth under conditions of full
sunlight, high leaf temperatures, and low nutrient availability, often characteristic of tropical
landscapes impacted by natural (Chambers et al., 2009) and human (Mesquita et al., 2001)
disturbances. Under optimal environmental conditions for photosynthesis, emissions of volatile
isoprenoids by leaves of many plant species can account for a few percent of Pn (Kesselmeier et
al., 2002). However, under stress conditions that diminish Pn but increase isoprene (Is) emissions
such as high leaf temperatures, emissions of Is can account for over 50% of Pn (Jardine et al.,
2014). While investments into Is production remains poorly understood among tropical plants
(Harley et al., 2004), the pattern of the photosynthetic carbon allocation has been discussed
through carbon-nutrient balance and growth-differentiation balance hypotheses (Stamp,
2004;Glynn et al., 2007;Massad et al., 2012). These hypotheses predicts the existence of trade-
off between investment in growth *versus* plant defense.
Thus, a hypothesis can be considered that fast growing pioneer tree species in secondary forests
do not produce volatile isoprenoids as secondary metabolites and instead dedicate these carbon
and energy resources to primary metabolites for enhanced biomass production and growth or
increased catabolism for energy generation during growth and maintenance respiration.
However, this hypothesis is not well supported in the literature as early successional pioneer
species have been observed with high volatile isoprenoid emission rates (Klinger et al.,
1998;Jardine et al., 2015).




An alternative hypothesis that is well supported in the literature is that investment of carbon and
energy resources into Is and monoterpene production and emissions by secondary forest species
protects photosynthesis during abiotic stress including high temperature stress, possibly through
antioxidant and energy/reducing equivalent consumption mechanisms (Vickers et al.,
2009a;Vickers et al., 2009b;Jardine et al., 2012b;Penuelas and Llusia, 2002;Grote et al.,
2014;Loreto and Velikova, 2001). Additional studies revealed possible connections between
volatile isoprenoid emissions and increased photorespiration during high leaf temperatures
(Jardine et al., 2014) and drought stress (Dani et al., 2014). Consistent with these potentially
important functional roles for pioneer species, a leaf and branch survey at four neo-tropical sites
(Harley et al., 2004) suggested that *Vismia guianensis* is an Is producer with a single leaf
emission value of 48 µg C $g^{-1}$ $hr^{-1}$ reported (or 11 nmol $m^{-2}$ $s^{-1}$ assuming a specific leaf area of 20
$m^2$ $kg^{-1}$) (Dias-Filho, 1995). A survey of Is emissions from tropical central Africa suggested that
Is emissions are higher in early successional forest communities relative to primary forests
(Klinger et al., 1998). In addition to Is, more recent field studies of *Cecropia sciadophylla* in the
Amazon suggest that not only can a fast growing tropical pioneer tree species emit volatile
isoprenoids, but that leaf emission rates of highly reactive monoterpenes such as cis and trans-β-
ocimene were among the highest yet observed from trees globally (Jardine et al., 2015).

Although volatile isoprenoid emissions are highly species specific, methanol emissions appear to
be a universal feature in plants attributed to the hydrolysis of cell wall methyl esters during
changes in cell wall chemical and physical properties (Fall, 2003;Fall and Benson, 1996). Leaf
methanol emissions have been shown to closely correlate with plant growth rates, especially at
the early stages of leaf development with young leaves consistently observed to be higher
emitters than mature leaves (Hüve et al., 2007;Nemecek-Marshall et al., 1995). In contrast, Is
emission capacity has been reported to increase considerably throughout leaf development in
close connection with photosynthetic capacity (Alves et al., 2014). However, phenological
controls on both methanol and Is emissions have not been studied together in tropical plants.

As a part of Observations and Modeling of the Green Ocean Amazon (GoAmazon 2014/5)
(Martin et al., 2015), we hypothesized that despite the high carbon and energy costs that could



otherwise be used for growth and maintenance, the highly abundant pantropical pioneer tree
species *V. guianensis* dedicates a significant fraction of recent photoassimilated carbon to
volatile isoprenoid emissions, due to their protective properties under abiotic stress. We further
hypothesized that this fraction changes as a function of light intensity due to the well-
documented light suppression of mitochondrial respiration at low light (e.g. the Kok effect)
which results in a large increase in Pn but only a relatively small increase in gross photosynthesis
(Sharp et al., 1984). In addition, recent mechanistic Is emission models suggest that during high-
light conditions where Pn is light-saturated, emissions of Is (and other volatile isoprenoids) may
continue to increase with increasing light due to increased excess available energy and reducing
power for the methylerythritol 4-phosphate (MEP) pathway (Grote et al., 2014). Consistent with
this model, a survey of tropical vegetation revealed strong light-dependent Is emission rates 2-3
times higher than those of temperate species (Lerdau and Keller, 1997). Moreover, tropical
leaves did not demonstrate a light-saturation in Is emissions which continued to increase with
light up to 2,500 mol $m^{-2}$ $s^{-1}$, the highest PAR fluxes studied (Lerdau and Keller, 1997). Thus,
although biochemical and modeling studies have identified mechanisms expected to cause
significant deviations between a constant linear ratio of volatile isoprenoid emissions and Pn as a
function of light, experimental observations in the tropics are extremely limited to investigate
these processes. Moreover, as laboratory studies have shown the classic uncoupling between net
photosynthesis and Is emissions occurs in tropical plants at high leaf temperatures, little *in situ*
information exists on this phenomenon in the tropics.

Here we first present new *in situ* observations during 2014 of leaf Is emissions and Pn as a
function of light intensity from *V. guianensis* in the central Amazon together with a reanalysis of
light-dependent monoterpene emissions from *C. sciadophylla* leaves in relation to Pn
(Jardine et al., 2015). Second, we present the results leaf Is emissions and Pn responses in *V.*
*guianensis* as a function of leaf temperature during 2015. Third, by taking advantage of the
rapidly developing leaves of *V. guianensis*, we also test the hypothesis that leaf phenology
differentially impacts methanol versus isoprenoid emissions. Finally, in order to further evaluate
the potential for secondary tropical forests to be important atmospheric sources of isoprene, we
present limited measurements of ambient daytime Is concentrations above a secondary rainforest



ecosystem in the central Amazon. We end by discussing the potential physiological roles of
volatile isoprenoids and methanol in secondary tropical rainforest ecosystems.

**2. Material and methods**

In this study, seven individuals of *Vismia guianensis* (Aubl.) Pers., a pioneer tree species from
the Hypericaceae family, were studied in the Reserva Biológica do Cuieiras (ZF2), a primary
rainforest biological reserve located approximately 60 km northwest of Manaus, in the central
Amazon Basin, Brazil. This reserve has an area of primary rainforest of roughly 230 km² and is
managed by the National Institute for Amazon Research (INPA). A nearby secondary rainforest
ecosystem (ZF3 reserve), located approximately 105 km northwest of Manaus, was also studied
for ambient concentrations of Is at the top of the canopy (~25 m) as a part of the biological
dynamics of forest fragments project (Gascon et al., 2001) (See **Fig. 1**).

**2.1 Ambient Concentrations of Is above the Secondary Forest Canopy**

Six ambient air thermal desorption tube samples (150 ml/min for 15 min) were sequentially
collected at the canopy height of 25 m on a walk up tower at the ZF3 site (coordinates: 02º
23'26.5" S and 59º53'0.7" W) on 23 April 2015 between 11:38-13:18. No samples were
collected on two additional thermal desorption tubes for background analysis. The thermal
desorption tubes were purchased commercially and filled with Quartz wool, Tenax TA, and
Carbograph 5TD adsorbents (Markes International, UK) and analyzed for Is concentrations using
a thermal desorption system interfaced with a gas chromatograph-mass spectrometer system
(GC-MS) at INPA in Manaus, Brazil as previously described (Jardine et al., 2014).

**2.2 Emission Responses to Light**

Emission responses to light for *V. guianensis* leaves under constant leaf temperature (30 °C) and
reference [$CO_2$] (400 ppm) were collected both in the field on intact branches and in the lab on
detached branches during July of 2014. Field observations of Is emissions and Pn for *V.*
*guianensis* leaves as a function of PAR intensity (0-2000 $\mu$mol m$^{-2}$ s$^{-1}$) under constant leaf
temperature (30 ºC) were based on the coupling of the LI-6400XT with a portable thermal
desorption tube sample collector as previously described (Jardine et al., 2015). Briefly, PAR
values of 0, 100, 250, 500, 1000, 2000 $\mu$mol m$^{-2}$ s$^{-1}$ were established for 10 min each with a





thermal desorption tube sample collected for each light level (75 ml min$^{-1}$ x 10 min). Blank tubes were also collected without a leaf in the enclosure at the beginning of the experiment when the light level was 0 µmol m$^{-2}$ s$^{-1}$. Is emissions were quantified using a thermal desorption GC-MS as previously described (Jardine et al., 2014). Emission responses to light were collected for two young and two young-mature leaves from intact branches in the field (one individual). In addition, light induced emission responses of detached branches (two individuals) were analyzed in the laboratory using PTR-MS. For these laboratory experiments, an additional light level of 3000 µmol m$^{-2}$ s$^{-1}$ PAR was included to evaluate the saturation of Pn and Is emissions at high light intensities. Both the GC-MS and PTR-MS systems were calibrated to Is using dynamic dilution of a commercial compressed gas standard (2.0 ppm Is, Apel-Riemer Environmental). Monthly GC-MS calibrations through a period that encompassed the July 2014 and 2015 field experiments (Nov 2013 - July 2015) demonstrated a high precision of Is quantitation by GC-MS; Is m/z 67 calibration slopes showed a relative standard deviation of 19.2%.

**2.3 Emission Responses to Temperature**

Emission responses to temperature for *V. guianensis* leaves under constant PAR (1000 µmol m$^{-2}$ s$^{-1}$) and reference [$CO_2$] (400 ppm) were collected on intact branches and in the field during July 2015. Field observations of Is emissions and Pn for *V. guianensis* leaves as a function of leaf temperature were conducted using the combined LI-6400XT/GC-MS system as described in section 2.2. Leaf temperatures (25, 27.5, 30.0, 32.5, 35, 37.5, and 40 ℃) were established for 10 min each with a thermal desorption tube sample collected for each temperature (50-75 ml min$^{-1}$ x 10 min). Blank tubes were also collected without a leaf in the enclosure at the initial temperature of 25 ℃ at the beginning of the experiment. Is emissions and Pn were quantified from 5 young-mature leaves from intact branches in the field (one leaf per individual).

**2.4 Pn combined with Is and methanol emissions as a function of leaf age in *V. guianensis***

In addition to being highly abundant in disturbed Amazon secondary forests (Mesquita, 2000), *V. guianensis* was selected because of its high leaf development rates, which produce two new apposing leaves roughly every month (G. Martins, personal communication). Leaves used were classified in three stages: young, young-mature, and mature. For all plants, young leaves occurred at the top of the branch or the first leaf stage, young-mature leaves occurring in the





second leaf stage, and mature leaves occurring in the third leaf stage. For each leaf age
experiment conducted during July 2014 in the field laboratory (six total individuals, one
experiment per day), large branches roughly 1 m tall were detached from the tree around noon
and immediately placed and recut in tap water before being transported to the field laboratory
and analyzed for gas exchange within 15 minutes of being cut. Upon arriving in the laboratory,
branches were placed under an LED plant growth light with between 300-600 µmol m$^{-2}$ s$^{-1}$
photosynthetically active radiation (PAR) at branch height. As air temperature of the laboratory
was roughly 25 ºC, the PAR and air temperature environment in the laboratory was likely lower
than the natural conditions under which the branch was removed. Leaf gas exchange
measurements were initiated by placing a young leaf (first leaf stage) in the enclosure of a
portable photosynthesis system (LI-6400XT, LI-COR Inc., USA) interfaced with a proton
transfer reaction – mass spectrometer (PTR-MS, Ionicon Analytik, Austria) as previously
described (Jardine et al., 2014). Is and methanol emissions were quantified using the mass to
charge ratios m/z 69 and 33 respectively using PTR-MS, while Pn, stomatal conductance, and
transpiration rates were quantified using the LI-6400XT. For each leaf age experiment, a single
young leaf (first leaf stage), young-mature leaf (second leaf stage), and mature leaf (third leaf
stage) were sequentially placed inside the chamber for 15 min each. Before and after each leaf
measurement, background measurements were collected for several minutes with an empty leaf
chamber. Thus, the time required for each leaf age experiment was roughly one hour beginning
around noon, during July of 2014. These leaf measurements were carried out under constant
PAR flux (1000 µmol m$^{-2}$ s$^{-1}$), leaf temperature (30 °C), and reference [$CO_2$] (400 ppm).

## 3 Results and Discussion

### 3.1 Ambient Concentrations of Is above the Secondary Forest Canopy

To evaluate for the first time the potential role of secondary forests as source of Is to the lower
tropical troposphere, we measured daytime ambient concentrations of Is at the top of a ~25 m
canopy in the ZF3 rainforest fragment site (Gascon et al., 2001). Daytime ambient Is
concentrations above secondary forest canopy at ZF3 where high (>10 ppb) and increased from
10.0 ppb at 11:38 to 10.9 ppb by 12:30. This was followed by a decreased to 10.5 ppb by 13:03,





possibly due to the reduction of light and temperature from afternoon cloud formation. As Is
concentrations from primary forests in the Amazon have been reported between 6-10 ppb
(Jardine et al., 2012b;Karl et al., 2009), these observations are consistent with the idea that
tropical secondary forests represent an important source of Is in the lower troposphere.

### 3.2 Emission Responses to Light

In order to investigate the possibility that the highly abundant pantropical pioneer species *V.*
*guianensis* dedicates a significant fraction of Pn to volatile isoprenoid emissions to the
atmosphere, we first conducted controlled light experiments on intact branches in the field using
a new portable photosynthesis and volatile organic compound emission system based on thermal
desorption GC-MS (Jardine et al., 2015). The results show that during the 2014 rainy season in
the central Amazon (13-May-2014), light-stimulation of Pn up to 15 $\mu$mol m$^{-2}$ s$^{-1}$ in young-
mature leaves were associated with Is emissions which continued to increase with light up to the
maximum PAR intensity (2000 $\mu$mol m$^{-2}$ s$^{-1}$) where emission rates were 30 nmol m$^{-2}$ s$^{-1}$. These
emission rates are higher than those reported by Harley *et al.*, 2004 of roughly 11 nmol m$^{-2}$ s$^{-1}$
from a *V. guianensis* leaf in the National Forest Tapajós, near Santarém, Brazil (Harley et al.,
2004). This emission rate is similar however, with Is emissions observed by PTR-MS in this
study during the 2014 dry season which ranged from 20-45 nmol m$^{-2}$ s$^{-1}$ under standard
conditions of PAR and leaf temperature, and up to 80 nmol m$^{-2}$ s$^{-1}$ at maximum PAR fluxes of
3000 $\mu$mol m$^{-2}$ s$^{-1}$ (see **Fig. 2**).

Although GC-MS results confirm that *V. guianensis* is a strong Is emitting species, Is collections
on thermal desorption tubes at each environmental light level represent the average emission rate
during each 10 min sample collection. Therefore, to analyze the relationship between Pn and Is
emissions as a function of PAR in greater temporal detail, real-time PTR-MS measurements of Is
emissions were collected simultaneously with real-time Pn measurements (**Fig. 2a**). The PTR-
MS system was installed in the field laboratory and detached branches of *V. guianensis* growing
just outside of the laboratory were utilized. Similar to the GC-MS measurements, *V. guianensis*
showed negligible Is emissions in the dark where Pn was negative (likely due to mitochondrial
respiration and the absence of photosynthesis). Moreover, upon first switching on the light, Pn





sharply increased from slightly negative in the dark to around 5.0 µmol m$^{-2}$ s$^{-1}$ at a PAR flux of
100 µmol m$^{-2}$ s$^{-1}$. With every increase in PAR up to the maximum of 3000 µmol m$^{-2}$ s$^{-1}$, Is
emissions continued to increase without any sign of saturation. In contrast, although Pn also
increased with PAR, it essentially saturated at PAR fluxes above 2000 µmol m$^{-2}$ s$^{-1}$. Thus, when
carbon flux emitted as Is was plotted against Pn, a strong non-linear relationship was observed
(**Fig. 2b**). As Pn increased with PAR, the fraction of Pn dedicated to Is emissions increased up to
1.9%. This non-linear effect could not be explained by an increase in leaf temperature as PAR
increased; throughout the range of PAR values, leaf temperatures remained between 30 +/- 1 ℃.

In light of the non-linear relationship between the fraction of Pn dedicated to Is emissions and Pn
for the pioneer species *V. guianensis*, we analyzed previously reported light-dependent
monoterpene data from the abundant pantropical pioneer species *C. sciadophylla* for a similar
non-linear relationship (Jardine et al., 2015). The results of the *C. sciadophylla* reanalysis also
revealed a strong non-linear relationship between the fraction of Pn emitted as monoterpenes and
Pn during controlled light experiments (graph not shown). The fraction of Pn dedicated to
monoterpene emissions continued to increase with PAR up to 1.9% at maximum PAR. Thus,
both *V. guianensis* and *C. sciadophylla* dedicate roughly 2% of Pn to volatile isoprenoid
emissions at 30 °C leaf temperature and show a strong increase in the fraction of Pn dedicated to
volatile isoprenoid emissions as PAR increases. These observations are consistent with a
growing body of evidence that the fraction of assimilated carbon transformed to volatile
isoprenoids increases with leaf energetic status (including high light and low atmospheric $CO_2$
concentrations) (Morfopoulos et al., 2014). While not captured by common Is emission
algorithms, the increased allocation of Pn to volatile isoprenoid emissions is captured by
energetic models of Is emissions (Morfopoulos et al., 2014). These observations imply that the
functional roles of volatile isoprenoids are particularly important under high light conditions and
could potentially be explained by a competition between photosynthesis and the MEP pathway
for adenosine triphosphate (ATP) and nicotinamide adenine dinucleotide phosphate (NADPH)
generated by the light reactions of photosynthesis (Grote et al., 2014).

Under low light conditions, the Benson-Calvin cycle dominates the consumption of ATP and
NADPH resulting in relatively large increases in Pn with a correspondingly small increase in



volatile isoprenoid production. In contrast, under light saturated conditions for Pn, excess ATP
and NADPH are consumed by the MEP pathway resulting in a relatively small increase in Pn
with a correspondingly large increase in volatile isoprenoid production. Finally, the Kok effect
may further contribute to this non-linear relationship at low light levels (Sharp et al., 1984). Low
PAR fluxes around the light compensation point for Pn have been shown to partially suppress
mitochondrial respiration which results in a relatively small increase in gross photosynthesis and
a correspondingly large increase in Pn. Thus, at low light levels, this would further contribute to
a relatively large increase in Pn with a correspondingly small increase in Is emissions.

**3.3 Emission Responses to Temperature**
A strong uncoupling of Is emissions and Pn was observed as a function of leaf temperature in
each of the *V. guianensis* leaves studied from 5 individuals (**Fig. 3**). Both Is emissions and Pn
increased together with leaf temperature from 25.0 up to 30.0-32.5 ℃. In contrast, at leaf
temperatures above 30.0-32.5 ℃, Is emissions continued to increase while Pn was strongly
suppressed, with 7% of Pn emitted as Is at 40 ℃. Therefore, distinct temperature optimum for Pn
(30.0-32.5 ℃) and Is (>40 ℃) exists for *V. guianensis* leaves. This classic uncoupling has been
shown to be influenced by the use of 'alternate' Is carbon sources including potential
extrachloroplastic substrates (Rosenstiel et al., 2004;Loreto et al., 2004;Karl et al., 2002) as well
as the re-assimilation of internally produced $CO_2$ (e.g. respiration, photorespiration) (Jardine et
al., 2014;Jardine et al., 2010). At the highest leaf temperature studied (40 ℃), 7% of the Pn on
average was emitted from *V. guianensis* leaves in the form of Is.

Interestingly, *V. guianensis* also produces large quantities of red latex as an herbivore deterrent
(Almeida-Cortez and Melo-de-Pinna, 2006). Although latex, or cis-polyisoprene, is produced
within the cytosol by the mevalonate pathway whereas Is is produced within the chloroplasts by
the MEP pathway, exchange of common intermediates such as isopentenyl diphosphate (IDP)
may occur (Chow et al., 2007). These data build on previous studies in Asia, which demonstrated
that latex producing trees can be strong emitters of volatile isoprenoids (Wang et al., 2007;Baker
et al., 2005).
**3.4 Pn combined with Is and methanol emissions as a function of leaf age in *V. guianensis***





A well-defined pattern of photosynthesis and Is /methanol emissions was observed according the
leaf age of *V. guianensis* as shown in real-time during two example leaf age experiments (**Fig. 4**)
and as an average of six leaf age experiments (**Fig. 5**). The results show that young leaves had
low Pn rates (2.7 +/- 2.2 $\mu$mol m$^{-2}$ s$^{-1}$) high methanol emissions (24 +/- 13 nmol m$^{-2}$ s$^{-1}$) but low
to undetectable Is emissions  (2 .1 +/- 0.2 nmol m$^{-2}$ s$^{-1}$). In contrast, young-mature leaves had
high Pn rates (12.9 +/- 5.2 $\mu$mol m$^{-2}$ s$^{-1}$) low methanol emissions (3.3 +/- 1.5 nmol m$^{-2}$ s$^{-1}$) but
high Is emissions (35.7 +/- 9.1 nmol m$^{-2}$ s$^{-1}$). In addition, mature leaves also showed the same
pattern as young-mature leaves with high Pn rates (10.6 +/- 5.2 $\mu$mol m$^{-2}$ s$^{-1}$) low methanol
emissions (3.0 +/- 1.1 nmol m$^{-2}$ s$^{-1}$) but high Is emissions (39.8 +/- 6.0 nmol m$^{-2}$ s$^{-1}$).

There is a wide range of morphological characteristic, chemical composition and physiological
activities of leaves depending on the developmental stage of the plant or tissue (Forrest and
Miller-Rushing, 2010;Richardson et al., 2013). In the case of photosynthesis, increased light
harvesting components, electron transport rates, and carboxylation efficiency occur in parallel
with the growth and development of leaves (Reich et al., 2009;Kikuzawa, 1995;Merilo et al.,
2009). The lower Pn rates observed in young *V. guianensis* leaves compared to mature leaves
(**Fig. 5**) is a pattern that is routinely observed in numerous studies (Alves et al., 2014;Reich et al.,
1991). The relationship between photosynthetic activity and leaf development can be explained,
in part, by the development of chloroplasts (Massad et al., 2012;Hikosaka, 2003). Previous
studies have confirmed increased levels of amino acids, proteins, nucleic acids and pigments
during leaf development, together with greater rates of carboxylation (Lohman et al.,
1994;Buchanan-Wollaston and Ainsworth, 1997;Egli and Schmid, 1999). Is from *V. guianensis*
leaves observed in this study also followed a similar developmental pattern with that of Pn rates.
This result is consistent with previous observations that in the early stages of leaf growth,
photoassimilates tend to be partitioned towards growth compounds at the expense of defense
compounds (Massad et al., 2012). This behavior can partially be explained by a limitation in
substrate for Is in young leaves because photosynthesis is one of the main processes responsible
for providing the required carbon intermediates, reducing equivalents, and ATP needed to
produce dimethylallyl pyrophosphate (DMAPP) required for Is biosynthesis (Loivamaki et al.,
2007;Sun et al., 2013). On the other hand, the demand for DMAPP is very high during leaf
expansion because this compound is essential for the synthesis of all plant isoprenoids including



photosynthetic pigments (Hannoufa and Hossain, 2012;Domonkos et al., 2013;Eisenreich et al., 2004;Opitz et al., 2014). Thus, due to the limited availability of DMAPP in young rapidly expanding leaves, a competition occurs for this substrate such that a larger fraction is allocated towards photosynthetic pigments (Rasulov et al., 2014). However, once the photosynthetic machinery is in place, a larger fraction of DMAPP may be dedicated to Is production and emissions (Rasulov et al., 2014). Once the photosynthetic machinery is established, high Is production rates may help protect against photoinhibition and photooxidation by consuming excess energy and reducing equivalents during conditions of light saturation for photosynthesis. Moreover, the antioxidant properties of Is have been well demonstrated (Jardine et al., 2012a;Vickers et al., 2009a;Loreto et al., 2001;Affek and Yakir, 2002). Is production lowers lipid peroxidation, quenches reactive oxygen species, and protects photosynthesis under oxidative stress (Loreto and Velikova, 2001). Previous studies have demonstrated the protective role of Is for photosynthesis during high leaf temperature stress (Sharkey et al., 2001) and a potential antioxidant mechanism was supported by the detection of Is oxidation products for high-temperature stress (Jardine et al., 2012a;Jardine et al., 2013).

In contrast to Pn and Is, emissions of methanol from *V. guianensis* leaves were eight times higher in young leaves than in mature leaves (**Fig. 5**), similar to patterns found in the literature from leaves of mid-latitude trees (Harley et al., 2007;Hüve et al., 2007;Nemecek-Marshall et al., 1995). Emissions of methanol have been closely associated with leaf growth rates (Hüve et al., 2007) caused primarily by the expansion of cell walls (Fall, 2003). As a consequence, young rapidly expanding leaves have consistently been observed to have higher methanol emissions than mature leaves (Hüve et al., 2007;Nemecek-Marshall et al., 1995). Methanol production is thought to be initiated during pectin demethylation reactions catalyzed by the enzyme pectin methylesterase (Bai et al., 2014). In this reaction, the hydrolysis of galacturonic acid methyl esters strengthens the cell wall while liberating methanol as a by-product (Harley et al., 2007;Bai et al., 2014;Hanson and Roje, 2001;Hüve et al., 2007). High rates of methanol emission may be associated with higher catalytic activity of this enzyme (Galbally and Kirstine, 2002;Hüve et al., 2007). In addition to growth processes, high methanol emissions have also been observed during stress and senescence processes (Cosgrove, 2005, 1999), possibly also mediated by pectin





demethylation reactions during physicochemical changes to cell walls. Thus, the high methanol
emissions from young leaves of *V. guianensis* may be due to both growth and stress processes.

**3.5 Potential Roles of Volatile Isoprenoids and Methanol in Secondary Tropical Ecosystems**
High emission rates of volatile isoprenoids have been observed from dominant central Amazon
pioneer species including *V. guianensis* (this study) and *C. sciadophylla* (Jardine et al., 2015)
as well early successional species in central Africa (Klinger et al., 1998). Although a
systematic survey of pioneer tree species in the tropics is needed, the potential for
widespread occurrence of Is emissions from secondary forest tree species is supported by the
single mid-day set of observations of high ambient Is concentrations (11.0 ppbv) above a diverse
secondary rainforest canopy in the central Amazon. The observations of increased Pn allocation
to volatile isoprenoid emissions as a function of light intensity provides additional support for a
functional role of volatile isoprenoid biosynthesis in minimizing photoinhibition by consuming
excess photosynthetic energy and reducing equivalents  (Morfopoulos et al., 2014) as well as
other potential direct and indirect antioxidant activities (Jardine et al., 2012a;Vickers et al.,
2009a;Velikova and Loreto, 2005).

The non-linear relationship between Is emissions and Pn in *V. guianensis* leaves is consistent
with a suppression of mitochondrial respiration at low light (Sharp et al., 1984) and an increased
dedication of photoassimilated carbon to Is biosynthesis via the methylerythritol 4-phosphate
(MEP) pathway under light saturating conditions of Pn, possibly due to the utilization of excess
available energy and reducing equivalents (Morfopoulos et al., 2014). These observations
suggest that volatile isoprenoids offer substantial protection to the photosynthetic machinery
against photoinhibition and oxidative damage under stress conditions such as the high-light and
leaf temperatures that are regularly experienced in secondary forest environments. In the case of
*V. guianensis*, we observed that Is emissions and photosynthesis rates increase together
throughout leaf development while methanol emissions decreased. Although not easily
distinguished in the present data set due to the low temporal resolution of leaf development
observations (roughly monthly resolved ages categorized into young, young-mature, and
mature), Pn has been shown to proceed Is emissions in young developing leaves by several days
to several weeks (Monson et al., 1994;Kuzma and Fall, 1993;Grinspoon et al., 1991). Methanol



emission patterns are also consistent to previous observations which have shown strong positive
relationships between leaf expansion rates and methanol emissions (Grinspoon et al., 1991).
Similar observations have been made at the ecosystem scale in a mixed hardwood forest in
northern Michigan, US during the spring growing season where a strong enhancement in
ecosystem emissions of methanol were observed together with an increase in leaf area index
(Karl et al., 2003).

**4. Conclusion**

The findings of this study show that abundant secondary rainforest tree species (e.g. *V.*
*guianensis* and *C. sciadophylla*) contribute high emissions of volatile isoprenoids to the
atmosphere that represent up to 2% of Pn under the standard leaf temperature of 30 ℃ and up to
7% under 40 ℃, the carbon and energy costs notwithstanding. Thus, the hypothesis that fast
growing pioneer tree species in secondary tropical forests do not produce volatile isoprenoids,
and instead dedicate these carbon and energy resources to enhanced growth and respiration
demands, is not supported.

High emission rates of volatile isoprenoids have been observed from dominant central Amazon
pioneer species including *V. guianensis* (this study) and *C. sciadophylla* (Jardine et al., 2015)
as well early successional species in central Africa (Klinger et al., 1998). Although a
systematic survey of pioneer tree species in the Amazon is needed, the potential for
widespread occurrence of Is emissions from secondary forest tree species is supported by the
single mid-day set of observations of high ambient Is concentrations (11.0 ppbv) above a diverse
secondary rainforest canopy in the central Amazon. The observations of increased Pn allocation
to volatile isoprenoid emissions as a function of light intensity provides additional support for a
functional role of volatile isoprenoid biosynthesis in minimizing photoinhibition by consuming
excess photosynthetic energy and reducing equivalents  (Morfopoulos et al., 2014) as well as
other potential direct and indirect antioxidant activities (Jardine et al., 2012a;Vickers et al.,
2009a;Velikova and Loreto, 2005). Together with previous studies, our observations support a
functional role for methanol production during cell wall expansion during growth (Fall, 2003)
and the establishment of photosynthetic machinery and a defense role for volatile isoprenoid



production to help protect this photosynthetic machinery against the abiotic stresses (Vickers et
al., 2009a) that are commonly experienced in secondary rainforest ecosystems.

**Acknowledgements**

The data used in this manuscript is available for download for research and educational purposes
at the following web link maintained by Lawrence Berkeley National Laboratory (LBNL, **link to**
**be included in final version**). We acknowledge the support from the Central Office of the Large
Scale Biosphere Atmosphere Experiment in Amazonia (LBA), the Instituto Nacional de
Pesquisas da Amazonia (INPA), and the Universidade do Estado do Amazonia (UEA). We
would like to expecially thank INPA researchers Giordane Martins and Ana Paula Florentino for
introducing our team to the fascinating science involving the fast growing pioneer species *Vismia*
*guianensis*. This research was supported by the GoAmazon 2014/5 and Next-Generation
Ecosystem Experiments (NGEE-Tropics) projects, which are funded by the Office of Biological
and Environmental Research of the U.S. Department of Energy (DOE), Office of Science,
through contract No. DE-AC02-05CH11231 to LBNL, as part of DOE's Terrestrial Ecosystem
Science Program.

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





**Figures**

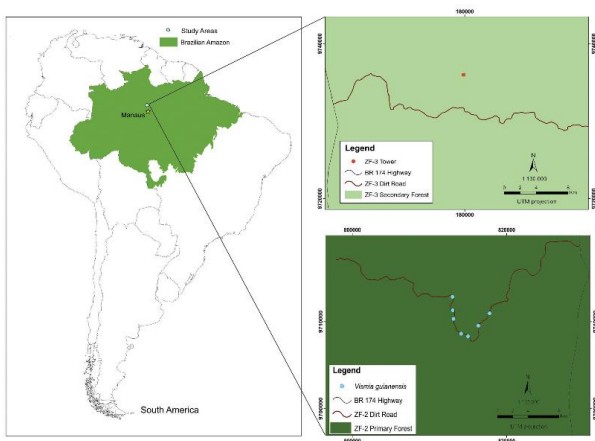


**Figure 1:** Location of the primary forest in the Reserva Biológica do Cuieras (ZF2) and the
secondary forest (ZF3) in the biological dynamics of forest fragments project near Manaus,
Brazil.



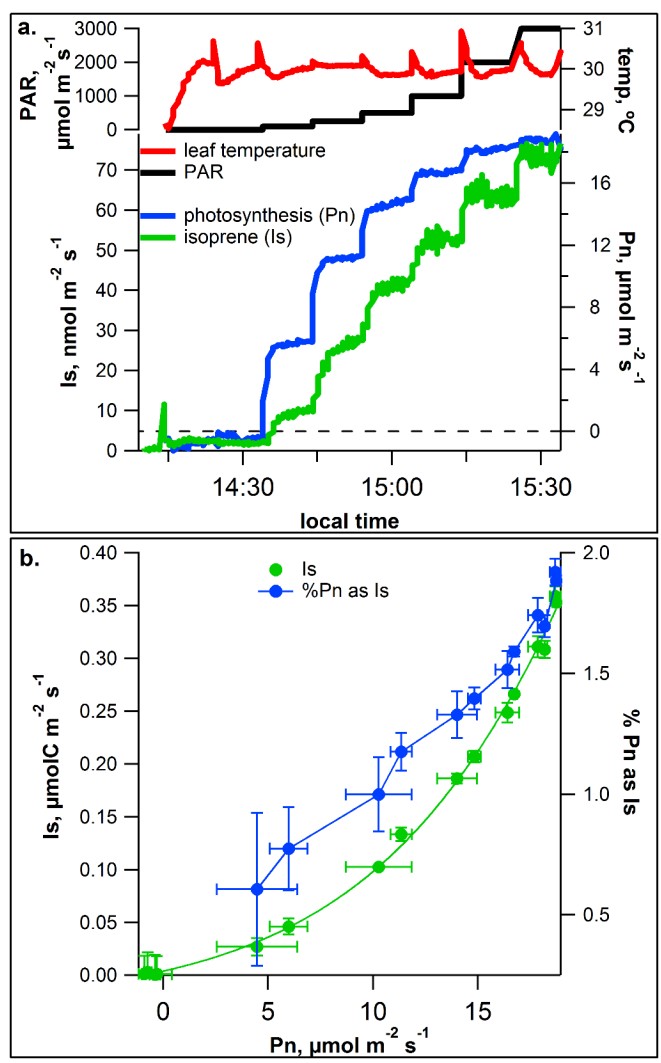


**Figure 2:** Real-time leaf net photosynthesis and isoprene emissions as a function of light intensity in the fast growing pantropical pioneer species *V. guianensis*. (**a**) Example time series plot of net photosynthesis (Pn), isoprene emission (Is) together with leaf temperature (temp) and photosynthetically active radiation (PAR) during a controlled light experiment under constant leaf temperature (30 ℃ +/- 1 ℃). (**b**) Isoprene emissions expressed in µmolC m$^{-2}$ s$^{-1}$ plotted against Pn (µmol m$^{-2}$ s$^{-1}$). Also shown is the % Pn dedicated to Is as a function of Pn. Note the increase in percentage of net photosynthesis emitted as isoprene as a function of Pn. Pn was determinate by a portable photosynthesis system (LI-6400XT, LI-COR Inc., USA) and Is by a proton transfer reaction – mass spectrometer (PTR-MS, Ionicon Analytik, Austria).

815

816



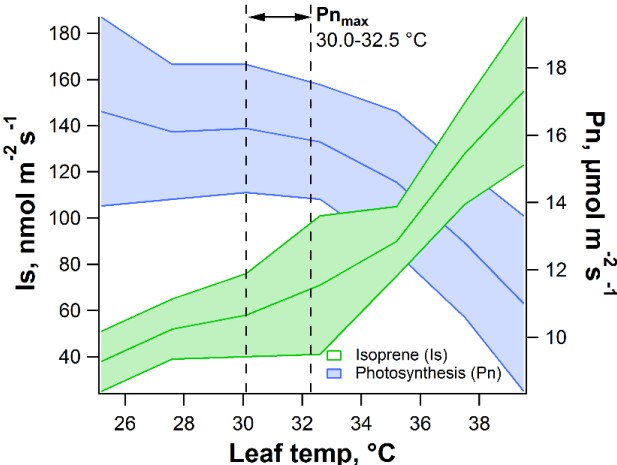

**Figure 3**: Average net photosynthesis (Pn) and isoprene emissions (Is) from *V. guianensis* leaves as a function of leaf temperature (average +/- 1 standard deviation, n = 5 leaves) under constant PAR of 1000 µmol m$^{-2}$ s$^{-1}$ and 400 ppm reference [$CO_2$]. Note the decline in Pn and the increase in Is with leaf temperature above 30.0-32.5 °C, where the majority of leaves showed an optimum in Pn. Pn was determinate with a portable photosynthesis system (LI-6400XT, LI-COR Inc., USA) and Is was determined using thermal desorption GC-MS.

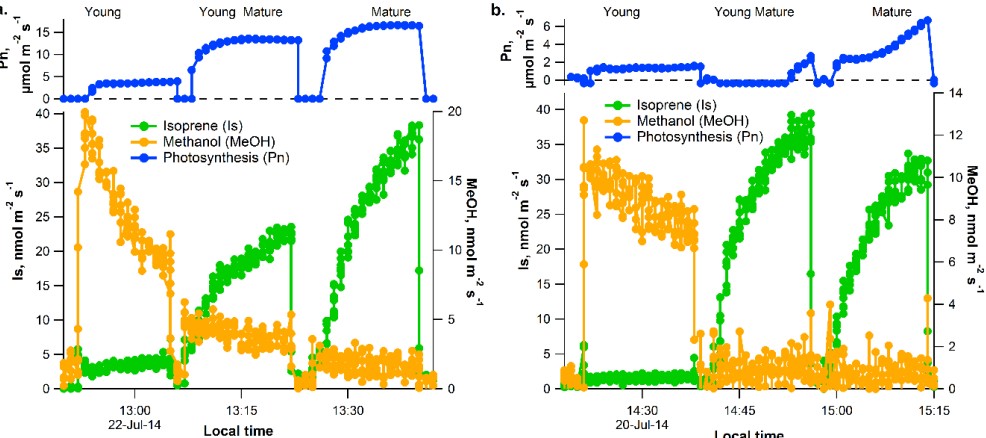

**Figure 4:** Example leaf age experiments of net photosynthesis (Pn, blue circles), methanol emissions (MeOH, orange circles) and isoprene emissions (Is, green circles) from two *V. guianensis* individuals. For each plant shown in **a.** and **b.** a young, young mature, and mature leaf were sequentially placed in the leaf enclosure for 15 min each after exposed to laboratory conditions following transport from the field. Throughout the leaf age experiments, PAR, leaf temperature and reference [$CO_2$] were held constant at 1000 µmol m$^{-2}$ s$^{-1}$, 30 °C and 400 ppm, respectively. Pn was determinate by a portable photosynthesis system (LI-6400XT, LI-COR Inc., USA). Is and MeOH by a proton transfer reaction – mass spectrometer (PTR-MS, Ionicon Analytik, Austria).




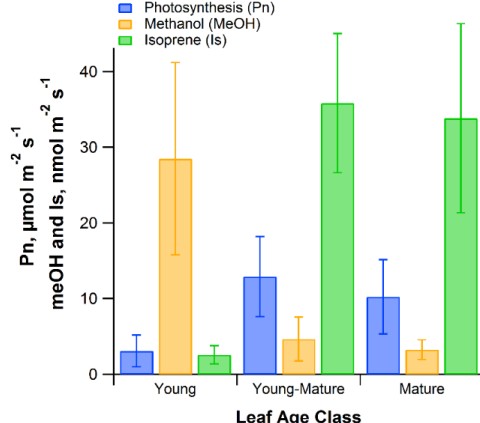


**Figure 5:** Average net photosynthesis rates (Pn, blue bars) together with leaf emissions of
Isoprene (Is, green bars) and methanol (MeOH, orange bars) for six *V. guianensis* individuals.
For each individual, volatile emissions and net photosynthesis rates were determined from
young, young-mature, and mature leaves. Error bars represent +/- one standard deviation (six
individuals, one branch per individual, one young, one young-mature, and one mature leaf per
branch). Pn was determinate by a portable photosynthesis system (LI-6400XT, LI-COR Inc.,
USA). Is and MeOH by a proton transfer reaction – mass spectrometer (PTR-MS, Ionicon
Analytik, Austria).