# Peer review of "Title: Methanol and Isoprene Emissions from the Fast Growing Tropical Pioneer Species Vismia"

_Atmospheric Chemistry and Physics, 2016_

## Referee Comment (RC1) · Anonymous Referee #1 · 9 Mar 2016

General Comment:

Jardine et al. conducted in situ observations of isoprene, methanol emissions from the pioneer species in central Amazon Forest. The major focus of this study was set on the emission responses to light and temperature as well as the change of emissions with respect to the leaf age. The experiment is well done, the manuscript is well structured and written, the reference list is conclusive and plots are clear. Thanks very much. There are two questions should be addressed before the final acceptance, which the paper deserves.

[Figure]

[Figure]

Minor comments:

1. As there is only a single-noon measurements of ambient isoprene concentration, why do authors believe the measurements supports "the potential for widespread occurrence of Is emissions from secondary forest tree species" in the Amazon? Were the measurements representative? What were the situation on the day of sampling (was it too hot, or any other stimulating factor existing)?

2. In line 322-323, "Both Is emissions and Pn increased together with leaf temperature . . .". But Fig.3 clearly shows a decrease of Pn. Could you please check.

---

## Referee Comment (RC2) · Anonymous Referee #2 · 18 Apr 2016

Jardine and coworkers show exciting research rich in hypotheses, useful observations and stimulating speculations to understand carbon allocation by tropical pioneer plant species as observed through field measurements. The work focuses on methanol and isoprene emission variance as a function of temperature, PAR and leaf age. This is an important issue as the forest regrows trying to compete for limited nutritional and energetic resources when it must necessarily be difficult to predict how the emissions will change due to anthropogenically influenced global change. The general lack of knowledge of VOC emissions from these neotropical species comes mainly from the unavailability of BVOC measurements in the field which is why more of such studies

are needed. However, similarly pioneering measurements were already done a decade ago by Harley et al. (2004) and the isoprene emission factors were consistent within a factor of four which points to the need of further explorations to understand the mechanisms behind plant's VOC emission and uptake.

More than anything, this reviewer appreciates how difficult it is to collect VOC data in the tropical rainforests and even for this reason the paper is strongly recommended for publication, almost as is. Thank you very much. It will certainly be a very useful contribution for the ACP community. While extremely enjoying the coherent story, I came up with just a few relatively very minor comments/suggestions which hopefully can inspire further discussion of this fascinating science direction and maybe some figures could be made even more clear.

General: 1) Why is the focus almost exclusively on methanol and isoprene? These are certainly extremely interesting and often most abundant BVOCs, but these plants must emit numerous other compounds such as stress tracers (e.g. temperature stress), higher terpenes, latex constituents, microbial VOCs, which could facilitate further understanding of issues such as uncoupling from Pn, photorespirations, biotic stress. I would be very surprised if these plants did not take up any of the VOCs to regenerate at least some carbon lost but this is not discussed.

2) Ideas that isoprene protects against temperature stress and that methanol is a growth-related BVOC are not very new hypotheses although perhaps still not perfectly supported. It seems to me that isoprene at least in part can just be a byproduct in the metabolism towards production of more specific compounds such as carotenoids, stress or microbially-induced monoterpenes such as b-ocimene. During stress, the requirement for production of larger stress molecules such as higher terpene antioxidants may be much larger potentially leading to higher emission of volatile byproducts. Thus, my question is if we can assume that a single compound such as isoprene or methanol plays a single, and non-complex role?

3) Further to previous comment, is there a reason why only a single role for methanol is seen? Methanol has numerous sources within plants, both emissions and deposition have been observed in ecosystem studies (e.g. Wohlfahrt et al., 2015). While it is well-known that methanol emissions are higher during plants' growth, other sources/sinks may be less known such as that it can be microbial substrate or a product of foliar microbiota.

4) Until very recently, the presence of microbes on leaf surfaces has been almost completely ignored in BVOC literature. This is shocking to me because there are ~10,000,000 microbes in 1 cm2 of phyllosphere (Lindow and Brandl, 2001) and they are not just hanging out there. In the tropics I would expect even higher densities of foliar microbes and they are known to be amazingly efficient chemical biolaboratories which require energy for multiplication. I think it could be relevant for this story (and other enclosure studies) at least giving a thought about the fact that epiphytic microbes interact with plants and recent studies clearly suggest that these microbes can significantly impact plant's metabolism (Peñuelas et al., 2014, Kanchiswamy, 2015). Example questions that remain to be answered are how microbial diversities change on the leaves during the rapid growth of pioneer species and if there is a shift in pectin decomposers which could explain methanol differences or if the microbes chew up on red latex to release additional source of isoprene to gain energy for division?

Specific

5) L. 106 "...possible connections between volatile isoprenoid emissions and increased photorespiration during high leaf temperatures". In your photorespiration hypothesis, do you account for microbial respiration?

6) Figure 2a.

- The slope of isoprene increase following PAR changes seems a little different at low PAR than at high PAR. This is a little surprising because isoprene is not sticky so I would expect almost instantaneous Is response. Can you exclude possibility of sticky

isoprene moiety from latex-conversion products at low PAR? At high PAR it becomes more clear that isoprene dominates the signal as the equilibration is much faster. Is it because it takes more time for the metabolic machinery to reach a steady state at low PAR than it is the case for high PAR?

- The figure suggests that isoprene emission in the dark is well above zero which is incredibly interesting so I wonder if it can explain the microbial and/or latex decomposition hypotheses.

7) Figure 4.

- Again, isoprene creeps up slowly (never reaching a steady state?) while much stickier methanol responds much faster. For quantification, did you trim out the unequilibrated portion or did you leave it in? How significant difference would this make? For the future studies I think it would make sense to suggest longer than 10 min sampling times to allow for full equilibration. It might be an instructive exercise to extract the fresh red latex from these plants and sniff with the PTRMS when heated to different temperatures. I might be wrong, but I would not be surprised if you saw some signal consistent with isoprene from these interesting poly-isoprene biopolymers.

- Further, if a and b denote different plants (the caption was not very clear to me) is it not surprising that isoprene emission at standard conditions is a factor of ∼2 higher in "young mature" leaf in b) than that in a)? It almost seems as if the "young mature" leaf was swapped with "mature" leaf in b) or is it the circadian rhythm of basal emission rates (e.g. Hewitt et al., 2011)? I also wonder why the "young mature" leaf in b emitted more isoprene at negative Pn? If Pn measurement worked well, and given the observed equilliration time, does this complete uncoupling suggest more like the isoprene moiety or conversion product from a different compound (possibly constituent of red latex)?

- Finally why does methanol show somewhat a logarithmic decay across all the samples in a) but less so in b)?

- Would you mind making the scale for methanol consistent in both panels?

8) Again, this study is extremely well done opening more doors to further hypotheses. In the conclusions, it would be nice to see suggestions for future work and further hypotheses that should be tested.

Technical: 9) L. 89 "These hypotheses predicts" should be "These hypotheses predict".

10) L. 433 should be "consistent with"

11) L. 598 Harley 2007, bgd. Did you mean to cite the discussion paper instead of the published version?

12) Figure 2, why is the PAR line ~15:20 inclined? If this is due to a gap in the sensor data it would be better to show the gap as NaN instead of interpolation.

13) Figure 5. This is a beautiful graph showing the essence of the story! For consistency, consider changing "meOH" to "MeOH".

References: Peñuelas, J., Farré-Armengol, G., Llusia, J., Gargallo-Garriga, A., Rico, L., Sardans, J., . . . Filella, I. (2014). Removal of floral microbiota reduces floral terpene emissions. Scientific Reports, 4, 6727. http://doi.org/10.1038/srep06727

Kanchiswamy, C. N., Malnoy, M., & Maffei, M. E. (2015). Chemical diversity of microbial volatiles and their potential for plant growth and productivity. Frontiers in Plant Science, 6, 151. http://doi.org/10.3389/fpls.2015.00151

Harley, P., Vasconcellos, P., Vierling, L., Pinheiro, C. C. d. S., Greenberg, J., Guenther, A., Klinger, L., Almeida, S. S. d., Neill, D., Baker, T., Phillips, O. and Malhi, Y. (2004), Variation in potential for isoprene emissions among Neotropical forest sites. Global Change Biology, 10: 630–650. doi:10.1111/j.1529-8817.2003.00760.x

Hewitt, C. N., Ashworth, K., Boynard, A., Guenther, A., Langford, B., MacKenzie, A. R., Misztal, P. K., Nemitz, E., Owen, S. M., Possell, M., Pugh, T. A. M., Ryan, A. C., and Wild, O. (2011). Ground-level ozone influenced by circadian control of isoprene

emissions, Nature Geosci, 4, 671-674.

Lindow, Steven E., and Maria T. Brandl. "Microbiology of the phyllosphere." Applied and environmental microbiology 69.4 (2003): 1875-1883.

Wohlfahrt, G., Amelynck, C., Ammann, C., Arneth, A., Bamberger, I., Goldstein, A. H., Gu, L., Guenther, A., Hansel, A., Heinesch, B., Holst, T., Hörtnagl, L., Karl, T., Laffineur, Q., Neftel, A., McKinney, K., Munger, J. W., Pallardy, S. G., Schade, G. W., Seco, R., and Schoon, N.: An ecosystem-scale perspective of the net land methanol flux: synthesis of micrometeorological flux measurements, Atmos. Chem. Phys., 15, 7413-7427, doi:10.5194/acp-15-7413-2015, 2015.
* * *

---

## Author Comment (AC1) · 22 Apr 2016

We greatly thank anonymous referee #1 for the time to review our manuscript and for the recognition that "The experiment is well done, the manuscript is well structured and written, the reference list is conclusive and plots are clear." The responses to the two minor comments are listed below.

Comment 1: As there is only a single-noon measurements of ambient isoprene concentration, why do authors believe the measurements supports "the potential for widespread occurrence of Is emissions from secondary forest tree species" in the Amazon? Were the measurements representative? What were the situation on the day of sampling (was it too hot, or any other stimulating factor existing)?

Response 2: The observations of ambient isoprene concentrations above the secondary forest are limited to mid-day on a single day (23 April 2015 between 11:38-13:18). However, six samples were sequentially collected during this period where full sun conditions persisted until 13:00 when clouds could be detected.

From Line 247: "Daytime ambient Is concentrations above secondary forest canopy at ZF3 where high (>10 ppb) and increased from 10.0 ppb at 11:38 to 10.9 ppb by 12:30. This was followed by a decreased to 10.5 ppb by 13:03, possibly due to the reduction of light and temperature from afternoon cloud formation."

Comment 2: In line 322-323, "Both Is emissions and Pn increased together with leaf temperature: : :". But Fig.3 clearly shows a decrease of Pn. Could you please check.

Response 2: Fig. 3 shows the average +/- 1 standard deviation of Pn for 5 leaves, one leaf per tree. We now clarify this point by rewriting this section;

Line 321: "A strong uncoupling of Is emissions and Pn was observed as a function of leaf temperature in each of the V. guianensis leaves studied from 5 individuals (Fig. 3). For 3 of the 5 leaves, Pn increased together with temperature and showed a clear optimum temperature of 30-32.5 °C and decreased at higher temperatures. The other two leaves showed decreases in Pn as temperatures increased above 25 °C. Thus, a relatively high standard deviation occurred at the lowest leaf temperature (25 °C) and a clear optimum in Pn between 30-32.5 °C was generally not observable from the average. Nonetheless, above 30-32.5°C, all 5 leaves showed a strong decrease in Pn.

In contrast, Is emissions from all 5 leaves increased with leaf temperatures above 25 °C; Is emissions continued to increase even while Pn was strongly suppressed up to the highest leaf temperatures studied (40 °C). Therefore, distinct temperature optima for Pn (30.0-32.5 °C) and Is (>40 °C) exists for V. guianensis leaves. This classic un-

coupling has been shown to be influenced by the use of 'alternate' Is carbon sources including potential extrachloroplastic substrates (Rosenstiel et al., 2004;Loreto et al., 2004;Karl et al., 2002) as well as the integration of photorespiratory substrates into the Calvin Cycle and the re-assimilation of internally produced CO2 (e.g. respiration, photorespiration) (Jardine et al., 2014;Jardine et al., 2010). At the highest leaf temperature studied (40 °C), 7% of the Pn on average was emitted from V. guianensis leaves in the form of Is."

---

## Author Comment (AC2) · 22 Apr 2016

We greatly thank anonymous referee #2 for the time to review our manuscript and for the recognition that the manuscript "shows exciting research rich in hypotheses, useful observations and stimulating speculations to understand carbon allocation by tropical pioneer plant species as observed through field measurements." The reviewer has stimulated many new ideas and possibilities for future research and has improved the manuscript. Responses to the comments and the changes made to the manuscript are listed below.

[Figure]

Comment 1: Why is the focus almost exclusively on methanol and isoprene? These are certainly extremely interesting and often most abundant BVOCs, but these plants must emit numerous other compounds such as stress tracers (e.g. temperature stress), higher terpenes, latex constituents, microbial VOCs, which could facilitate further understanding of issues such as uncoupling from Pn, photorespirations, biotic stress. I would be very surprised if these plants did not take up any of the VOCs to regenerate at least some carbon lost but this is not discussed.

Response 1: Isoprene and methanol were the most abundant VOCs emitted by the pioneer species that we could quantify based on calibration standards in the field using PTR-MS and GC-MS. Indeed a plethora of other volatile plant/microbial compounds may be emitted under high temperature and light stress including green leaf volatiles, benzenoids, and nitriles, and lipid peroxidation products that would most certainly provide additional physiological constraints on metabolic responses to environmental change. However, this is beyond the scope of the present manuscript which focused not only on characterizing light, temperature, and leaf developmental emission response curves, but mechanistically linking them to developmental and physiological processes associated with growth and carbon assimilation. It should be noted here that leaf emissions of higher terpenoids including mono and sesquiterpenes were not detected by the analytical systems under the conditions used.

Comment 2: Ideas that isoprene protects against temperature stress and that methanol is a growth-related BVOC are not very new hypotheses although perhaps still not perfectly supported. It seems to me that isoprene at least in part can just be a byproduct in the metabolism towards production of more specific compounds such as carotenoids, stress or microbially-induced monoterpenes such as b-ocimene. During stress, the requirement for production of larger stress molecules such as higher terpene antioxidants may be much larger potentially leading to higher emission of volatile byproducts. Thus, my question is if we can assume that a single compound such as isoprene or methanol plays a single, and non-complex role?

Response 2: Isoprene protection of photosynthesis under environmental extremes is not a new hypothesis, and as discussed, at least four mechanisms are currently being discussed in the literature. In this manuscript, we do not view isoprene as a byproduct of the MEP pathway with other more important compounds such as carotenoids and pigments the main product. As described in the manuscript, the initial hypothesis that fast growing pioneer species in secondary tropical forests allocate carbon primarily to growth at the expense of isoprene defenses is rejected. We describe four functional roles of isoprene biosynthesis previously discussed in the literature that may be critical for supporting photosynthesis in tropical pioneer species including 1) Minimization of ROS formation by acting as a sink for excess photosynthetic energy and reducing equivalents, 2) Thermally stabilizing membranes, 3) Stabilizing membranes and photosynthetic machinery by directly reacting with ROS and other radicals, and 4) Signaling mechanisms involving isoprene oxidation products. None of these mechanisms have been well evaluated in the tropics as most of the studies on isoprene functional protection of photosynthesis have occurred in temperate plant species. Here, we provide new and rare data on isoprene emissions in tropical pioneer species, which sheds new light into these mechanisms in the tropics. In particular, while energetic isoprene emission models predict a non-linear relationship between isoprene emissions and Pn (and therefore support the first mechanism), few observations of this phenomenon exist globally. Our data adds new clear experimental evidence to support such an energetic model and therefore has important implications for both the physiological understanding of isoprene in the tropics as well as global emission modeling efforts aimed at quantifying its role in the Earth System.

As described in the manuscript, although methanol emissions have been linked to plant growth and development in numerous studies, observations in the tropics are lacking. Moreover, the developmental controls of both isoprene and methanol together and their relationship with the establishment of photosynthesis in the tropical pioneer species provides new insights into physiological functioning. The results suggests that methanol is involved in the establishment of the photosynthetic machinery while isoprene is involved in protecting this machinery once established.

Comment 3: Further to previous comment, is there a reason why only a single role for methanol is seen? Methanol has numerous sources within plants, both emissions and deposition have been observed in ecosystem studies (e.g. Wohlfahrt et al., 2015). While it is wellknown that methanol emissions are higher during plants' growth, other sources/sinks may be less known such as that it can be microbial substrate or a product of foliar microbiota.

Response 3: This point is well taken. We agree that methanol may not only be produced by plants but may also be consumed by plants. This is the subject we have studied extensively and is the topic of a future manuscript from our research group. We note here that our experimental methods did not allow for quantification of bidirectional methanol exchange between leaves and the atmosphere as methanol-free air was fed into the leaf chamber, thereby eliminating the possibility of net uptake fluxes. Rather than trying to upscale leaf level measurements to ecosystem scale (which must take into account bidirectional exchange), our goal was to evaluate the role of leaf developmental stages on methanol emissions in connection with isoprene emissions and net photosynthesis rates.

Comment 4: Until very recently, the presence of microbes on leaf surfaces has been almost completely ignored in BVOC literature. This is shocking to me because there are _10,000,000 microbes in 1 cm2 of phyllosphere (Lindow and Brandl, 2001) and they are not just hanging out there. In the tropics I would expect even higher densities of foliar microbes and they are known to be amazingly efficient chemical biolaboratories which require energy for multiplication. I think it could be relevant for this story (and other enclosure studies) at least giving a thought about the fact that epiphytic microbes interact with plants and recent studies clearly suggest that these microbes can significantly impact plant's metabolism (Peñuelas et al., 2014, Kanchiswamy, 2015). Example questions that remain to be answered are how microbial diversities change on the leaves during the rapid growth of pioneer species and if there is a shift in pectin

decomposers which could explain methanol differences or if the microbes chew up on red latex to release additional source of isoprene to gain energy for division?

Response 4: It is indeed fascinating to consider the role of microbial metabolism on the leaf-atmosphere fluxes of isoprene and methanol. Allthough we do not have microbial data, these possibilities are real and need to be evaluated, especially in the warm and moist tropics where a multitude of microbial niches exists, particularly in the understory where reduced light and increased moisture availability supports a rich ecosystem of microbes on leaf surfaces and even within plant cells. The comments are particularly interesting given that isoprene consumption by soil microbes has been demonstrated and methylotrophs may be highly abundant on plants where they may use the plant-derived methanol as a carbon and energy source. However, it should be noted that the environments where pioneer tree species grow are extremely hot, dry, and exposed to high levels of UV radiation which may act to sterilize microbes on surfaces. Nonetheless, plant-microbial interactions deserve a great deal more attention, particularly in primary rainforests, but are beyond the scope of the current study.

Comment 5: L. 106 ": : :possible connections between volatile isoprenoid emissions and increased photorespiration during high leaf temperatures". In your photorespiration hypothesis, do you account for microbial respiration?

Response 5: See response to comment 4 regarding microbial metabolism.

Comment 6: Figure 2a. - The slope of isoprene increase following PAR changes seems a little different at low PAR than at high PAR. This is a little surprising because isoprene is not sticky so I would expect almost instantaneous Is response. Can you exclude possibility of sticky isoprene moiety from latex-conversion products at low PAR? At high PAR it becomes more clear that isoprene dominates the signal as the equilibration is much faster. Is it because it takes more time for the metabolic machinery to reach a steady state at low PAR than it is the case for high PAR? - The figure suggests that isoprene emission in the dark is well above zero which is incredibly interesting so I

wonder if it can explain the microbial and/or latex decomposition hypotheses.

Response 6: The slope of isoprene increase following PAR change is different at low PAR than at high PAR and is the basis for the isoprene energy hypothesis as described in the manuscript. The possibility of sticky isoprene at low PAR is ruled out, steady state isoprene emissions are reached within minutes of changing PAR intensities but it does appear to be the case that steady state is reached faster at high PAR than at low PAR. This may be due to stomatal effects which open in response to changing light. Nonetheless, only steady state isoprene emission values are used in Fig. 2b. Isoprene emissions in the dark are very low and previous studies have shown that, other than after 5-10 minutes where residual isoprene biosynthesis may occur, light is required for isoprene emissions. Nonetheless, microbial latex decomposition is an interesting hypothesis (and commercially important!).

Comment 7: Figure 4. - Again, isoprene creeps up slowly (never reaching a steady state?) while much stickier methanol responds much faster. For quantification, did you trim out the unequilibrated portion or did you leave it in? How significant difference would this make? For the future studies I think it would make sense to suggest longer than 10 min sampling times to allow for full equilibration. It might be an instructive exercise to extract the fresh red latex from these plants and sniff with the PTRMS when heated to different temperatures. I might be wrong, but I would not be surprised if you saw some signal consistent with isoprene from these interesting poly-isoprene biopolymers.

Response 7: For the leaf age emission studies, cut branches were removed from the plant in the field and returned to the laboratory. The slow response of isoprene emissions to reach steady state is a result of stomatal conductance initially being very low and increasing during the measurments for some leaves. However, only the highest emission rates were used for figure 5 for both methanol and isoprene. The decreasing methanol emissions with time on some of the leaf samples may be due to both a stress of placing the leaf in the chamber, but also to the opening of the stomata with time

(methanol gas-aqueous phase partitioning).

Comment 8- Further, if a and b denote different plants (the caption was not very clear to me) is it not surprising that isoprene emission at standard conditions is a factor of 2 higher in "young mature" leaf in b) than that in a)? It almost seems as if the "young mature" leaf was swapped with "mature" leaf in b) or is it the circadian rhythm of basal emission rates (e.g. Hewitt et al., 2011)? I also wonder why the "young mature" leaf in b emitted more isoprene at negative Pn? If Pn measurement worked well, and given the observed equilibration time, does this complete uncoupling suggest more like the isoprene moiety or conversion product from a different compound (possibly constituent of red latex)?

Response 8: Yes a and b denote different plants as described in the figure caption. While considerable variability existed in the methanol and isoprene emissions among the young, young-mature, and mature leaves from the different plants studied, a significant difference was observed between young and young-mature/mature (Fig. 5). The emission of isoprene from the young-mature leaf in Fig. 5b with negative Pn is discussed in the paper as the result of closed stomata initially, followed by a later re-opening leading to positive Pn values. This can be understood in terms of alternate carbon sources for isoprene as described in the manuscript.

Comment 9: - Finally why does methanol show somewhat a logarithmic decay across all the samples in a) but less so in b)? - Would you mind making the scale for methanol consistent in both panels?

Response 9: As described on line 406: "In addition to growth processes, high methanol emissions have also been observed during stress and senescence processes (Cosgrove, 2005, 1999), possibly also mediated by pectin demethylation reactions during physicochemical changes to cell walls. Thus, the high methanol emissions from young leaves of V. guianensis may be due to both growth and stress processes." We prefer to keep the scale for methanol full-scale for both plants in Fig. 4. A common scale for

methanol emissions from all plants is used in Fig. 5.

Comment 10: Again, this study is extremely well done opening more doors to further hypotheses. In the conclusions, it would be nice to see suggestions for future work and further hypotheses that should be tested.

Response 10: We prefer to leave future work and suggestions out of the manuscript. This would however, be ideal for a review paper or a proposal.

Comment 11: Technical: L. 89 "These hypotheses predicts" should be "These hypotheses predict". Response 11: This correction has been made.

Comment 12: L. 433 should be "consistent with" Response 12: This correction has been made.

Comment 13: L. 598 Harley 2007, bgd. Did you mean to cite the discussion paper instead of the published version? Response 13: The published version is now cited.

Comment 14: Figure 2, why is the PAR line _15:20 inclined? If this is due to a gap in the sensor data it would be better to show the gap as NaN instead of interpolation.

Response 14: This is not a data gap nor an interpolation. The PAR source was set to its absolute maximum of 3,000 $\mu$mol m-2 s-1 and the setpoint took longer to establish than the lower setpoints.

Comment 15: Figure 5. This is a beautiful graph showing the essence of the story! For consistency, consider changing "meOH" to "MeOH".

Response 15: "meOH" was changed to "MeOH"
* * *

---

## Author Response (AR2)

09 May 2016

Dear Dr. Tuukka Petäjä,

On behalf of the coauthors, please find the updated research article, manuscript number **acp-2016-53,** entitled "Methanol and Isoprene Emissions from the Fast Growing Tropical Pioneer Species *Vismia guianensis* (Aubl.) Pers. (Hypericaceae) in the Amazon Basin" for consideration as a publication in the ACP special issue, "Observations and Modeling of the Green Ocean Amazon (GoAmazon2014/5)". Following the online per review in the Discussion format, we greatly appreciate your technical comments, which has helped improve the readability of the document for final publication in ACP. Details of the responses to your comments and the changes made to the manuscript are listed below.

Sincerely,

Kolby Jardine (on behalf of all coauthors)
Research Scientist
Climate and Ecosystem Sciences Division
Lawrence Berkeley National Laboratory

**Comment #1:** Figure caption 2. Line 816: Note the….is not a complete sentence. Please complete. ….determined….

**Response #1:** This sentence has been reworded as follows, "Also shown is the increase in %Pn dedicated to Is as a function of Pn."

**Comment #2**: "determined" spelled incorrectly in figure captions 2-5.

**Response #2**: 'determined' is now spelled correctly in the figure captions 2-5.